# Thermodynamic Aspects of Ion Exchange Properties of Bio-Resins from Phosphorylated Cellulose Fibers

**DOI:** 10.3390/polym17152022

**Published:** 2025-07-24

**Authors:** Lahbib Abenghal, Adrien Ratier, Hamid Lamoudan, Dan Belosinschi, François Brouillette

**Affiliations:** 1Innovation Institute in Ecomaterials, Ecoprodcuts and Ecoenergies, Biomass-Based (I2E3), Biochemistry, Chemistry, Physics and Forensic Science Department, Université du Québec à Trois-Rivières, 3351 boul. des Forges, Trois-Rivières, QC G9A 5H7, Canada; adrien.ratier87@hotmail.fr (A.R.); francois.brouillette@uqtr.ca (F.B.); 2CTT Group, 3000 Av. Boullé, Saint-Hyacinthe, QC J2S 1H9, Canada; hlamoudan@gcttg.com

**Keywords:** cellulose, phosphorylation, adsorption, ion exchange, thermodynamic, heavy metals, wastewater

## Abstract

Phosphorylated cellulose is proposed as a bio-resin for the removal of heavy metals, as a substitute for synthetic polymer-based materials. Phosphorylation is carried out using kraft pulp fibers as the cellulose source, with phosphate esters and urea as reactants to prevent significant fiber degradation. Herein, phosphorylated fibers, with three types of counterions (sodium, ammonium, or hydrogen), are used in adsorption trials involving four individual metals: nickel, copper, cadmium, and lead. The Langmuir isotherm model is applied to determine the maximum adsorption capacities at four different temperatures (10, 20, 30, and 50 °C), enabling the calculation of the Gibbs free energy (*ΔG*), entropy (*ΔS*), and enthalpy (*ΔH*) of adsorption. The results show that the adsorption capacity of phosphorylated fibers is equal or even higher than that of commercially available resins (1.7–2.9 vs. 2.4–2.6 mmol/g). However, the nature of the phosphate counterion plays an important role in the adsorption capacity, with the alkaline form showing a superior ion exchange capacity than the hybrid form and acid form (2.7–2.9 vs. 2.3–2.7 vs. 1.7–2.5 mmol/g). The thermodynamic analysis indicates the spontaneous (*ΔG* = (-)16–(-)30 kJ/mol) and endothermic nature of the adsorption process with positive changes in enthalpy (0.45–15.47 kJ/mol) and entropy (0.07–0.14 kJ/mol·K). These results confirm the high potential of phosphorylated lignocellulosic fibers for ion exchange applications, such as the removal of heavy metals from process or wastewaters.

## 1. Introduction

Over the last few decades, industrialization has expanded globally, contributing to the development of many processes, products, and devices which are essential to daily human life. This rapid growth has also contributed significantly to environmental pollution, including the release of hazardous metals, greenhouse gases (such as carbon dioxide and methane), and plastics [1]. Consequently, the demand for raw materials such as metals (iron, copper, lead, zinc, nickel, etc.) is growing exponentially, promoting new metal extraction activities in several countries. Some of these metals are called heavy metals because of their density of more than 5 g/cm^3^. This category of metals contributes significantly to water and soil pollution, with estimates indicating that among more than 10 million contaminated sites, more than 50% are affected by heavy metals and/or metalloids [2]. Even at low concentrations, they can integrate into the food chain, posing a serious threat to animal and human health, when concentrations exceed acceptable levels. Various industries are looking for materials that can be used to remove or recover heavy metals from recycled water or wastewater before discharging them into the environment to reduce pollution [3,4].

Traditionally, chemical precipitation was used to remove heavy metals from wastewater. This method involves the use of an alkaline solution to precipitate heavy metals, followed by filtration or other separation processes [5,6]. Despite the effectiveness of chemical precipitation, the resulting sludge is a costly to treat and hazardous residue. Therefore, other methods can be used to treat wastewater such as reverse osmosis, ultrafiltration, adsorption, biosorption, phytoremediation, and electrodialysis [7,8,9]. Nowadays, ion exchange is the method of choice at the industrial scale to recover and remove heavy metals. Ion exchange resins are characterized by their porous structure, thermal resistance (up to 100 °C), and stability over a wide pH range (1–14) [10]. Ion exchange occurs through functional groups attached to the resin’s skeleton. It leads to the electrostatic capture of the metal and the release of the counterion into the solution. There are two groups of ion exchange resins: anion exchange resins, with carboxylic or sulfonic acid functional groups, and cation exchange resins with amino functional groups (–NH_2_ or –NR_2_) [10,11,12]. Synthetic ion exchange resins are the most widely available on the market, thanks to their low cost and high total exchange capacity, sometimes exceeding 4.5 mmol/g [13]. Numerous studies have dealt with the use of synthetic resins. For example, S. Edebali et al. used Amberlite IRA96 and Dowex 1 × 8 to remove Cr (VI) from aqueous solution and found that maximum adsorption capacities were 0.46 mmol/g for Amberlite IRA96 and 0.54 mmol/g for Dowex 1 × 8 [14]. X. Liu et al. used D301R and DEX-Cr resins to evaluate Cr (VI) removal efficiency. Results showed that at pH 3, the adsorption capacity was 4.77 mmol/g for DEX-Cr and 3.73 mmol/g for D301R after 80 min of contact [15]. However, these materials present a host of drawbacks that limit their use, such as non-biodegradability, intensive energy consumption, dependence on fossil fuels, health risks, and recycling challenges.

Recently, biopolymers have been studied as good alternatives to synthetic ion exchange resins due to their abundance, biodegradability, recyclability, non-toxicity, and low cost. For instance, R. Boughanmi et al. used a mixture of pectin (with a negative charge) and chitosan (with a positive charge) to create polyelectrolyte complexes for the removal of heavy metal ions such as Cd, Ni, Fe, Mn, and Cu [16]. X. Zhao et al. employed a one-step method to prepare a chitosan- and lignin-based hydrogel for ion exchange applications. The prepared hydrogel gave maximum adsorption capacities of 0.67 and 1.55 mmol/g for Pb and Cu, respectively [17]. X. Rong et al. chemically modified a starch surface with tris(ethylenediamino-N-ethoxy) titanate isopropoxy to add amine groups to the starch skeleton to enhance its removal capacity [18]. However, chitosan, pectin, and starch have certain limitations, the most important of which are their poor stability under acidic conditions and the difficulties associated with material recycling after treatment, making metal recovery impractical. Fortunately, cellulose fibers can exhibit excellent ion exchange capacity after chemical modification, and recovery of the final material is much easier than chitosan or starch due to their fiber length (2–3 mm for softwood and 0.7–1.5 mm for hardwood) and stability under highly acidic conditions [19]. The phosphorylation of cellulose is one of the reactions used to add a phosphate function bearing a high negative charge that can be used as a cation exchange resin. The chemicals most commonly used in this reaction are phosphoric acid, phosphate salts, or phosphate esters in combination with a base such as pyridine, N,N-dimethylformamide, or urea [20,21]. The phosphate groups added to the cellulose skeleton provide a high charge density with counterions (such as sodium, ammonium, or hydrogen), which can be replaced by metal ions in solution (Figure 1). Numerous studies have demonstrated the effectiveness of these fibers in recovering heavy metals via an ion exchange mechanism [22]. For example, T. Oshima et al. used phosphorylated bacterial cellulose as a bio-resin for lanthanide ion removal [23]. A. M. A. Nada et al. used phosphorylated cotton stalks with a sodium ion exchange capacity of 244 meq to eliminate heavy metals such as Cr, Cu, Co, Ni, Cd, and Pb [24]. A. Mautner et al. used phosphorylated nanocellulose papers to create a membrane for copper elimination [25]. In a previous study, electrospun chitosan combined with phosphorylated nanocellulose was used as a biosorbent for the removal of cadmium ions. The maximum adsorption capacity obtained using the Langmuir model was 232.55 mg/g at 25 °C, demonstrating the high affinity of cadmium ions for amine groups (from chitosan) and phosphate groups (from phosphorylated nanocellulose) [26]. However, the thermodynamics of this system have never been studied before and are essential for a better understanding of the ion exchange process.

In this study, the thermodynamic aspects of the ion exchange system from phosphorylated cellulose were studied. The bio-based ion exchange resin was synthesized following a procedure developed by our research group. The method uses phosphate esters instead of phosphoric acid as phosphorylation reagent and achieves chemical modification of cellulose fibers without significant evidence of degradation. Four heavy metals were used individually to build model process water for the ion exchange process, namely nickel, copper, cadmium, and lead. The Langmuir isotherm model was applied to determine the maximum adsorption capacity, the adsorption constant, and the corresponding change in Gibbs free energy (*ΔG*). The experiments were reproduced at four different temperatures (10, 20, 30, and 40 °C), allowing for the calculation of the thermodynamic entropy (*ΔS*) and enthalpy (*ΔH*) of adsorption.

## 2. Materials and Methods

### 2.1. Materials

Bleached softwood kraft pulp fibers (KF) were supplied by Kruger Wayagamack Inc. (Trois-Rivières, QC, Canada). The phosphorylated fibers were obtained using phosphate esters prepared in our laboratory. All chemical reagents were used as received from different suppliers: 1-decanol, phosphorus pentoxide (P_2_O_5_), copper (II) sulfate pentahydrate (CuSO_4_.5H_2_O), cadmium (II) sulfate (CdSO_4_), nickel (II) sulfate (NiSO_4_), lead (II) nitrate (Pb(NO_3_)_2_) (Sigma-Aldrich, ACS reagent ≥ 99.0%, St. Louis, MI, USA), and urea (Alfa Aesar, Ward Hill, MA, USA). Dowex 50WX2-400 and Dowex Marathon C commercial resins were obtained from Sigma-Aldrich and used without further purification.

### 2.2. Synthesis of Phosphate Esters and Phosphorylated Cellulosic Fibers

The phosphate ester (PE) was synthesized following methods previously described by L. Abenghal et al. and Y. Shi et al. [27,28]. The process involves the combination of a fatty alcohol (1-decanol) with phosphorus pentoxide (P_2_O_5_) and water (to convert P_2_O_5_ into phosphoric acid (H_3_PO_4_)) (Figure 2). The reaction was conducted under continuous stirring at 60–80 °C for 8 h. The resulting PE was used without further purification. The obtained PE consists of a mixture of phosphate monoesters (predominant) and a smaller proportion of phosphate diesters.

Kraft pulp fibers were phosphorylated using the synthesized PE and molten urea (Figure 3), following the method described by H. Lamoudan et al. and Y. Shi et al. [27,29]. The molar ratio of the PE:urea:anhydroglucose unit was set at 3:17:1, and the reaction was carried out in an oven at 150 °C for 3 h under a light vacuum. At the end of the reaction, the fibers were washed multiple times to remove any side products and unreacted reagents with deionized water and finally rinsed with denatured alcohol. The reaction conditions yield fibers with grafted phosphates carrying an ammonium ion and a proton as counterions, as shown in Figure 3 and Figure 4. This specific chemistry is referred to as hybrid form (KFP-HYB) in the present study. The sodium form is obtained after two successive treatments of the phosphorylated fibers as described in Figure 4. First, the phosphorylated fibers were treated with 0.1 N HCl in excess, and then with 0.1 N NaOH in excess. Finally, the fibers were thoroughly rinsed with demineralized water to eliminate any excess of NaOH, to obtain a sodium form referred to as KFP-Na (Figure 4).

### 2.3. Fiber Characterization

***Fiber length distribution:*** A Lorentzen and Wettre Fiber Tester Plus fiber quality analyzer (FQA) was used to evaluate kraft fiber dimensional changes after the phosphorylation reaction. In total, 0.5 g of dried fibers was dispersed in 500 mL of deionized water using a high-shear mixer (Silverson L4, East Longmeadow, MA, USA) for 5 min at 3500 rpm. The fiber length, width, and length-weighted percentage of fines were determined for each sample. Five measurements were carried out for each sample to achieve greater precision.

***Total charge:*** The total charge of cellulose fibers, before and after phosphorylation, was determined by conductometric titration, as described in our previous work [30]. The sample preparation for titration involved three steps: first, all samples were protonated with 0.1 N aqueous HCl for 30 min and then dried in an oven at 60 °C. Next, 1 to 2 g of dried fibers were dispersed in 600 mL of deionized water, and 1 mL of sodium chloride solution (0.5 N) was added to stabilize conductivity. The final solution was titrated with 0.1 N sodium hydroxide using an automatic titrator (Dosimat 765; Metrohm Brinkmann, Camden, NJ, USA) and a Thermo Orion conductivity meter (model 150; Thermo Fisher Scientific, Waltham, MA, USA).

***Surface charge:*** The surface charge was measured using two standard 0.1N polyelectrolyte solutions: polydiallyldimethylammonium chloride (poly-DADMAC, positively charged) and poly(vinyl sulfate) potassium salt (PVSK, negatively charged), following the method described by H. Lamoudan et al. [31]. In total, 0.2–0.4 g of fibers was dispersed in 50 mL of deionized water, followed by the addition of 10 mL of 0.1 N polyDADMAC. The mixture was stirred at room temperature for 30 min. Next, the slurry was filtered using a 202-grade filter paper and transferred to a 100 mL flask. Finally, 10 mL of the filtrate solution was titrated with PVSK to measure the excess polyDADMAC. The analysis was conducted with a Mutek PCD03 analyzer (BTG Americas, Norcross, GA, USA) equipped with an automatic titrator. Calculating the ratio between the poly-DADMAC consumed by the surface of fibers and the weight of the dried fibers allows for the determination of the amount of surface charge on the fibers.

***Zeta potential:*** Zeta potential was measured to evaluate the mobility of the fiber’s anionic charge due to grafted phosphate groups. A sample of 10 to 15 g of fibers was dispersed in 700 mL of deionized water, and a sodium chloride solution (1 N) was used to adjust the conductivity of the suspensions to a range of 0.1–8 mS/cm. The analysis was performed using a Mutek SZP-06 analyzer (BTG Americas, Norcross, GA, USA).

***FT-IR Spectroscopy:*** FTIR spectroscopy was used to identify characteristic phosphate vibration bands. The analysis was performed using an iS10 FTIR spectrometer (Thermo Fisher Scientific, Waltham, MA, USA), with a spectral range of 4000 to 400 cm^−1^.

***Morphological Examination:*** Scanning electron microscopy (15 kV, variable pressure), coupled with energy-dispersive X-ray spectroscopy (SEM/EDX, Hitachi High-Tech Corporation, Tokyo, Japan, SU1510 with Oxford X-max 20 mm^2^), was used to assess morphological changes as well as to evaluate the changes in the elemental composition of cellulose fiber surface through phosphorylation.

### 2.4. Adsorption/Ion Exchange Trials

***Preparation of cation metal solutions:*** Stock solutions were prepared by dissolving an appropriate amount of copper sulfate (1000 mg/L Cu^2+^), nickel nitrate (1000 mg/L Ni^2+^), cadmium nitrate (1500 mg/L Cd^2+^), and lead nitrate (2000 mg/L Pb^2+^) in deionized water using a 2000 mL volumetric flask for greater precision. A total of 6 working solutions were obtained by diluting this stock solution to perform the adsorption trials.

***Adsorption tests:*** Batch adsorption tests were used to assess the ability of KFP to remove heavy metals such as Pb, Ni, Cd, and Cu from prepared solutions at four different temperatures: 10, 20, 30, and 50 °C. Prior to each test, the adsorption cell containing 100 mL of metal ion solution was thermostated externally for 30 min to reach equilibrium with a Neslab RTE-111 heater/cooler system. Then, a mass of 0.2 g of dry KFP fibers was added to the model water containing heavy metal and stirred for 30 min to reach the adsorption equilibrium at the specific temperature. The sample was quickly filtered on a G2 glass filter to separate fibers from the aqueous solution.

***Analysis of metal ion concentration***: The initial (c_0_) and final (c_e_) cation concentrations in aqueous solutions were determined by complexometric titration. Colorimetric titration with standard 0.1 N ethylenediaminetetraacetic acid (EDTA) was used to determine the concentration of ions in the solutions. Two indicators were used: murexide for nickel and copper ions, which changes in color from yellow to pink, and Eriochrome black T for cadmium and lead ions, which changes in color from mauve to blue.

***Isotherm model of adsorption:*** Given that adsorption in the case of KFP fibers is mainly dictated by the ion exchange capacity of the phosphate moiety grafted on the surface of the cellulose fibers, only the Langmuir monolayer model of adsorption (Equation (1)) was considered in the present study. In addition, only the results for the treatment carried out at 20 °C are presented for convenience [32].(1)$$\mathrm{Langmuir}:\;q_{e}=\frac{q_{max}K_{L}C_{e}}{1+K_{L}C_{e}}\;\;\mathrm{or}\;\;\frac{C_{e}}{q_{e}}=\frac{1}{K_{L}q_{max}}+\frac{C_{e}}{q_{max}}$$
where

*q_e_*—represents the adsorption capacity at equilibrium (mg/g);*C_e_*—is the (final) equilibrium concentration of metal in solution (mg/L);*K_L_*—is the Langmuir constant (L/mg);*q_max_*—is the maximum adsorption capacity (mg/g).

The slope and the intercept of linear Equation (1) allow for calculating the Langmuir constant (*K_L_*) and the maximum adsorption capacity (*q_max_*).

The amount of metal ions adsorbed onto the KFP surface at equilibrium (*q_e_*) was calculated using Equation (2):(2)$$\mathrm{Adsorbed}\;\mathrm{amount}:\;q_{e}=\;\frac{\left(C_{0}-C_{e}\right)\;V_{sol}}{m_{F}\;(1-TH)}$$
where

*V_sol_*—is the volume of ion metal solution used in trial (L);*m_F_*—is the weight of the KFP fiber used in trial (g);*TH*—is the moisture content of the KFP fiber, expressed as units;*C_0_*—is the initial concentration of metal in solution (mg/L);*C_e_*—is the (final) equilibrium concentration of metal in solution (mg/L).

***Thermodynamics of adsorption:*** The Langmuir constant allows for calculating the change in the Gibbs free energy of adsorption at a specific temperature, as shown in Equation (3):(3)$$\Delta G=\;-R\;T\;ln(\frac{K_{L}}{M})$$
where

*ΔG*—is the change in Gibbs free energy of the system at a temperature *T* (kJ/mol);*R*—is the gas constant (kJ/mol·K);*T*—is the temperature of the system (K);*K_L_*—is the Langmuir constant (L/mg);*M*—is the atomic mass of the metal (g/mol).

Additionally, the change in the entropy (*ΔS*) and enthalpy (*ΔH*) of the system can be determined if the adsorption is evaluated at different temperatures. The graphical representation of the Gibbs free energy of adsorption (*ΔG*) at different temperatures allows for drawing a straight line through the experimental points according to Equation (4). The slope and the intercept correspond to changes in the thermodynamic entropy (*ΔS*) and enthalpy (*ΔH*) of the adsorption system, respectively.(4)ΔG=ΔH−TΔS
where

*ΔG*—is the change in the Gibbs free energy of the system at a temperature T (kJ/mol);*T*—is the temperature of the system (K);*ΔH*—is the change in the enthalpy of the system (kJ/mol);*ΔS*—is the change in the entropy of the system (kJ/mol·K).

## 3. Results and Discussion

### 3.1. Fiber Characteristics

The characterization of the basic properties of fibers before and after chemical modification is essential to predict their behavior in various applications. The grafting of phosphate groups onto the cellulose backbone results in various modifications to fiber morphology, behavior in solution, and chemical structure. The morphology of KFP was evaluated using SEM and compared with KF as a reference. As shown in Figure 5, the morphology of the fibers was not significantly affected by the reaction conditions, with KF and KFP fibers having similar dimensions. The density of fractures on the fiber surface increased moderately after phosphorylation. KFP appears to be shredded in some areas. The presence of these fractures is more than likely due to the acidic environment in which the reaction takes place. However, no fibrils are present on the surface of KFP, with the surface showing a similar smoothness to KF. EDX analysis (bottom of Figure 6) confirmed successful phosphorylation, with the phosphorus content ranging from 3.8 to 4.5 atomic % for all KFP forms. The nitrogen content of KFP-HYB is 5.5% due to the presence of phosphate groups associated with ammonium (NH_4_^+^) as counterion. This percentage significantly decreases to 2.9% in KFP-H, as ammonium is replaced by protons (H^+^). Finally, KFP-Na exhibits a low nitrogen content of 0.3%, indicating that ammonium has been completely replaced by sodium whose content increases to 6.4%.

Phosphate is mainly grafted to cellulose as phosphate moiety, leading to various vibrational bands in IR spectroscopy, such as P=O, P-O-C, P-O-H, O-P-O(NH_4_^+^). Therefore, the FTIR spectrum (Figure 7) was used to qualitatively confirm the presence of phosphate in the structure of the fibers after the phosphorylation reaction. The first change observed in the FTIR spectrum is the significant decrease in the intensity of the peak at 3350 cm^−1^, corresponding to the stretching of cellulosic OH. This decrease is a clear indication of the substitution of OH with phosphate groups in KF after the phosphorylation reaction. The presence of phosphate can also be confirmed by bands located at 1270, 1010, and 950 cm^−1^, corresponding to P=O, P-O-C, and P-O-H, respectively. As expected from previous studies [33], the aliphatic chain of phosphate esters is absent in all KFP spectra as it undergoes hydrolysis in the reaction environment. The peaks at 1664 cm^−1^ and 2360 cm^−1^ correspond to water and carbon dioxide, respectively. KFP-Na shows a peak at 2360 cm^−1^, which is more intense than the other forms, due to traces of NaOH reacting with CO_2_ to form NaHCO_3_. The fiber’s polarity increases after the grafting of phosphate groups, which leads to an increase in the equilibrium humidity content in all forms of KFP compared to KF. Acid and alkaline treatments lead to slight changes in the FTIR spectra of KFP. The intensity of the peak at 3350 cm^−1^ for KFP-HYB is much lower than for the KFP-H and KFP-Na forms, due to the presence of ammonium as a counterion. Moreover, the presence of a high-intensity peak at 1435 cm^−1^ in KFP-HYB confirms the presence of ammonium phosphate.

The KF fibers used in this study are derived from softwood, with an average length generally ranging from 1 to 3 mm depending on the process conditions. However, exposure to high temperatures, low pH levels, or swelling reagents such as urea can result in the hydrolysis or degradation of the cellulose fibers, thereby reducing their average length. As anticipated, the fiber length of KF decreased slightly from 2.14 mm to 1.95 mm (Table 1) after the phosphorylation reaction. The use of PE instead of phosphoric acid as phosphorylation reagent is advantageous due to its lower acidity, which helps minimize the impact on fiber length during the reaction. In this regard, Shi et al. reported that the acidity of the medium is the main reason for fiber degradation. They found that fibers phosphorylated using phosphate esters had an average length of 1.34 mm, whereas those treated with phosphoric acid measured only 0.334 mm in length [27]. They attributed this to the higher acidity of phosphoric acid (pKa_1_ = 2.15) compared to that of the phosphate ester (pKa_1_ = 3.01). Similarly, Ablouh et al. observed that the phosphorylation with phosphoric acid reduced the average length of KF fibers by 50% after the reaction [33]. However, the main drawback of using PEs is their surfactant-like behavior, which often creates challenges during the washing process due to the foam generation. The fibers need to be washed multiple times to remove ester residues, leading to a reduction in the fine content from 0.7 to 0.2% (Table 1). In practice, a high percentage of fines is undesirable as they tend to accumulate in the process water, potentially causing operational issues. Finally, the average fiber width increased from 26.6 to 32.1 µm, which can be attributed to the presence of phosphate onto the surface of cellulose. Phosphate is a very polar moiety, which causes an additional swelling of the cellulose fiber wall. Consequently, the phosphate groups increase the hydrophilicity of the fibers, resulting in a higher retention of water. The moisture content at equilibrium increased from 4% for KF to 8% for KFP on average.

The electrostatic properties of KF significantly changed after the phosphorylation reaction. As shown in Table 1, the total charge of KF increases from 80 to 4850 mmol/kg, and the surface charge increases from 30 to 530 µeq/g by phosphorylation. In general, the grafted phosphate moiety on cellulose contains two hydroxyl groups with different pKa. The first hydroxyl group is highly acidic and dissociates easily in water, while the second is less acidic and thus less prone to dissociation. This is clearly highlighted in Figure 8, where the conductometric titration curves of KF and PKF are shown. The titration curve of KFP exhibits two equivalence points. Until the first equivalence point (zone 1), sodium hydroxide neutralizes the dissociated proton in solution, while between the first and second equivalence point (zone 2), sodium hydroxide neutralizes the undissociated proton on the fiber. Equivalent points are detected by linear regression of each of the three zones of the conductimetric titration curve obtained.

The value of the Zeta potential is of great importance in an ion exchange process. The dissociation of counterions from the grafted phosphate leads to an electronegatively charged surface of KFP fibers in water. This generates an electric double-layer capacitor with a Zeta potential that can be measured with a streaming potential device. The degree of dissociation of counterions from the grafted phosphate and the electrostatic interaction between them establish the Zeta potential value. A good ion exchange material must have a Zeta potential close to zero, indicating a propensity to efficiently capture and tightly hold metal ions from wastewater. Table 1 shows the Zeta potential values of KFP and KF. The KFP has a Zeta potential between −25 to −8 mV compared to −60 to −20 mV for KF (Figure 9). These closer to zero values for KFP indicate that the counterions of grafted phosphate are mainly concentrated in the compact Stern layer, whereas for KF, they are more concentrated in the diffuse layer. In addition, the charge density of KFP is around 60 times higher than that of KF, exhibiting the remarkable potential of KFP as bio-resin for ion exchange applications.

### 3.2. Adsorption Results

#### 3.2.1. Langmuir Model of Adsorption

Three adsorption models, Langmuir, Freundlich, and Dubinin–Radushkevich, were applied to the results to identify the most suitable one for this study. The determination coefficient (R^2^) was calculated for each model, with the Langmuir model consistently demonstrating the best fit across all tested metals and fiber types. Therefore, the linear Langmuir model was applied to all adsorption trials. Linear regressions are shown in Figure 10, and the determination coefficients (R^2^) are presented in Table 2. The Langmuir model is in good agreement with experimental data for heavy metal cation absorption on KFP. With a few exceptions, the coefficient of determination is greater than 0.99, regardless of the species of heavy metal exchanged or the type of counterion present on KFP. Consequently, the adsorption can be considered mostly directed to active sites represented by the grafted phosphate moiety, forming a monolayer of heavy metals on the KFP surface.

Preliminary tests to determine the contact time needed to reach the equilibrium of adsorption were carried out. In this regard, two heavy metals, one with a relatively low atomic weight (nickel) and the other representing a high atomic weight metal (lead), have been analyzed. The results of these experiments are shown in Figure 11. The equilibrium of adsorption is reached in the first two minutes of contact between the fibers and the metal ion solutions, highlighting a very high rate of ion exchange as required in practical industrial applications. Despite the adsorption equilibrium being reached in such a short period of time, the isotherms of adsorption were conducted over 30 min periods to ensure precision and confidence in the thermodynamic study.

#### 3.2.2. Maximum Adsorption Capacity

The maximum ion exchange capacity of KFP was calculated using the Langmuir model for all isotherm trials, and the results are presented in Table 3. KFP appears to be less selective to the species of the heavy metal divalent cation present in solution. However, the nature of the counterion present on the grafted phosphate of KFP significantly influences the adsorption capacity. As shown in Table 3, the maximum adsorption capacity of heavy metal ions for different KFP forms increases in the following order: KFP-H < KFP-HYB < KFP-Na, with values ranging from 1.7 to 2.9 mmol/g. The adsorption process is less favorable when the proton (H^+^) is used as a counterion, while sodium (Na^+^) is the most favorable. The most plausible explanation is given by the weak dissociation level of the second proton of KFP-H, as confirmed by conductometric titration (see Figure 8). This proton has a tighter bond to KFP, hindering the ion exchange and therefore reducing the value of the maximum adsorption capacity. The values obtained with phosphorylated fibers containing different counterions were compared with those of commercial resins such as Dowex 50WX2-400 and Dowex Marathon C. The results revealed that the maximum adsorption achieved with the commercial resins was 2.6 mmol/g, which is very similar to the maximum adsorption obtained with KFP.

Another observation is the correlation between the number of active sites of adsorption on KFP and the maximum adsorption capacity. The total charge of KFP is around 4.8 mmol/g, which is equivalent to a theoretical adsorption capacity of 2.4 mmol/g of heavy metal divalent cation from the solution. The maximum adsorption capacity calculated with the Langmuir model is less than the theoretical value in the case of KFP-H, similar for KFP-HYB, and is slightly higher for KFP-Na. This behavior can again be attributed to the mobility of the counterion present on the grafted phosphate of KFP. Sodium offers the highest mobility of all the counterions used in the study, and thus, the value of the maximum adsorption capacity is the highest in this case.

Finally, the maximum adsorption capacity of KFP was compared to some commercially available resins. The results in Table 3 show an ion exchange capacity slightly inferior in the case of the KFP-H form, equal in the case of the KFP-HYB form, and even superior in the case of the KFP-Na form, to that of commercial resins. As a result, KFP proves to be a greener technological solution, competing with synthetic commercial ion exchange resins.

A comparison with other bio-based adsorbents derived from cellulose fibers is necessary to highlight the significance of this study and the effectiveness of phosphorylated cellulose as a bio-resin for removing heavy metals. As shown in Table 4, the maximum adsorption capacity of phosphorylated cellulose used in this study is higher than that of most raw materials commonly employed as bio-adsorbents for heavy metals. The cellulose derivatives most often used are in the form of microfibers or nanofibers, which present several drawbacks, such as high production costs, the use of hazardous chemicals, a complex preparation process, and difficult recovery after adsorption. In most cases, the reuse and recyclability of the material after adsorption are impossible, potentially creating new environmental problems. The KFP bio-resins mitigate most of these issues.

#### 3.2.3. Thermodynamics of Adsorption

To determine the optimal conditions for the adsorption process, it is essential to assess the thermodynamics of the complex process governing the nature of the interaction between the adsorbate and the adsorbent. Adsorption experiments were carried out at various temperatures to investigate the thermodynamics of the system. First, the change in Gibbs free energy (*ΔG*) of the system was calculated using Equation (3). Negative *ΔG* values in the −16 to −30 kJ/mol range were obtained regardless of adsorption temperature, adsorbent species, or KFP form, indicating in all cases the spontaneous nature of the adsorption process.

The linear regression analysis of *ΔG* against *T* leads to curves with negative slopes (Figure 12). The coefficient of determination (R^2^) is in all cases above 0.95, confirming a very good model prediction of the thermodynamics of adsorption for KFP. The slope and the intercept of the linear curves in Figure 12 represent the change in entropy (*ΔS*) and enthalpy (*ΔH*) of the adsorption system. These values are highlighted in Table 5. Both calculated enthalpy and entropy are positive in all cases, confirming the endothermic nature of the adsorption process on KFP. The negative slopes of the linear curves shown in Figure 12 confirm once again this statement as *ΔG* values decrease when the temperature increases. From a thermodynamic standpoint, a spontaneous and endothermic process can only occur if the system disorder increases, i.e., the entropy is higher after adsorption on KFP. In other words, the driving force of adsorption for KFP is the increase in the entropy of the system. It seems that the surface of KFP is more ordered before adsorption, hosting evenly distributed and more mobile counterions like H^+^, NH_4_^+^, and Na^+^. After adsorption, the degree of disorder increases because the surface mainly hosts heavy metal cations and a small fraction of initial counterions acting as defects in this pattern.

## 4. Conclusions

A new bio-based ion exchanger, featuring phosphate moieties grafted onto cellulose fibers (KFP), was successfully synthesized and characterized. Phosphate esters, used as the phosphorylation agent, imparted a high negative charge to the fibers (4850 mmol/kg), a zeta potential ranging from –25 to –8 mV, and a fiber length of 1.95 mm, indicating that the fiber structure remained intact. The anionic phosphate groups capture hazardous metal ions from aqueous solutions via an ion exchange mechanism, releasing non-hazardous counterions such as sodium, ammonium, or hydrogen in return. This study shows that the adsorption capacity of phosphorylated fibers depends on the type of counterion associated with the phosphate groups: 2.7–2.9 mmol/g for the sodium form, 2.3–2.7 mmol/g for the hybrid form, and 1.7–2.5 mmol/g for the acid form. The adsorption process is spontaneous (Δ*G* = −16 to −30 kJ/mol) and endothermic, with enthalpy values ranging from 0.45 to 15.47 kJ/mol and entropy values from 0.07 to 0.14 kJ/mol·K. The increase in the system entropy is the main driving force behind the adsorption process with KFP. These results highlight the potential of KFP for the industrial-scale treatment of hazardous wastewater, especially those containing heavy metals, which are commonly generated by manufacturing plants worldwide.

## Figures and Tables

**Figure 1 polymers-17-02022-f001:**
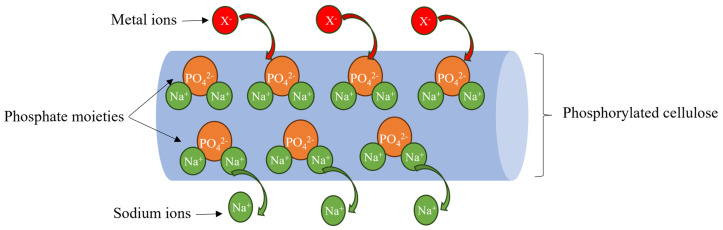
Ion exchange mechanism of phosphorylated cellulose for heavy metal removal.

**Figure 2 polymers-17-02022-f002:**
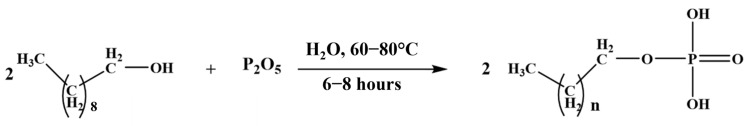
Phosphate esters synthesis using phosphorus pentoxide.

**Figure 3 polymers-17-02022-f003:**
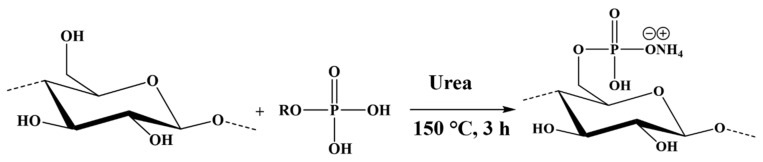
Phosphorylation of KF with phosphate esters and urea (fibers are obtained in their hybrid form).

**Figure 4 polymers-17-02022-f004:**
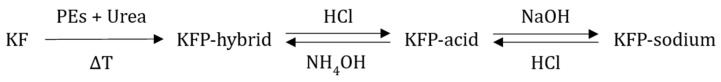
Interchanging the counterions of grafted phosphate to produce various forms of phosphorylated kraft fibers.

**Figure 5 polymers-17-02022-f005:**
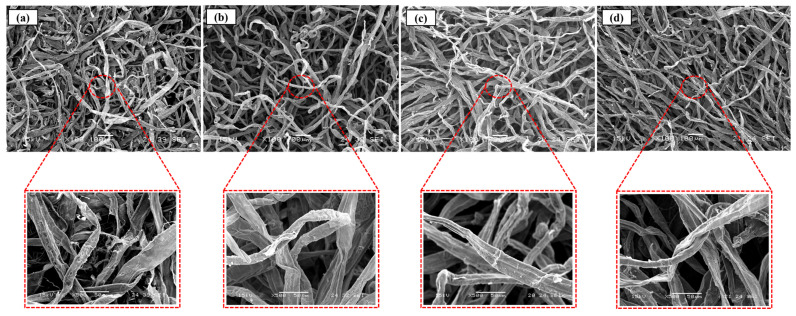
SEM analysis of KF (**a**), KFP-HYB (**b**), KFP-H (**c**), and KFP-Na (**d**).

**Figure 6 polymers-17-02022-f006:**
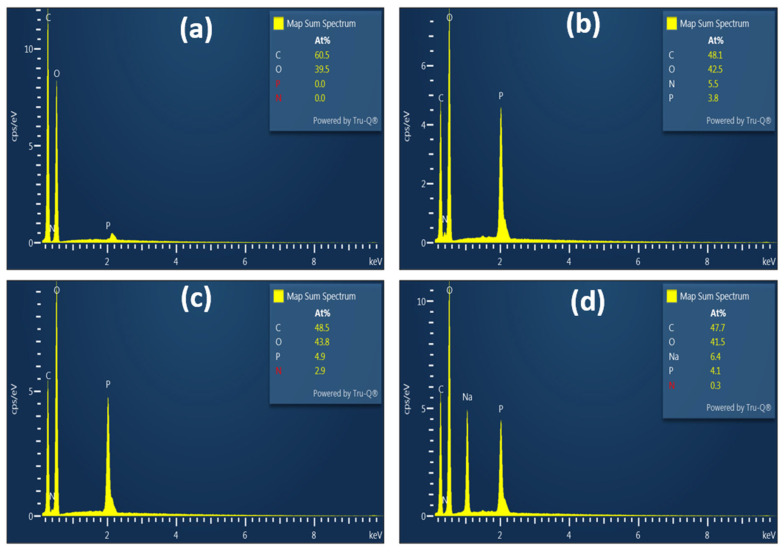
EDX analysis of KF (**a**), KFP-HYB (**b**), KFP-H (**c**), and KFP-Na (**d**).

**Figure 7 polymers-17-02022-f007:**
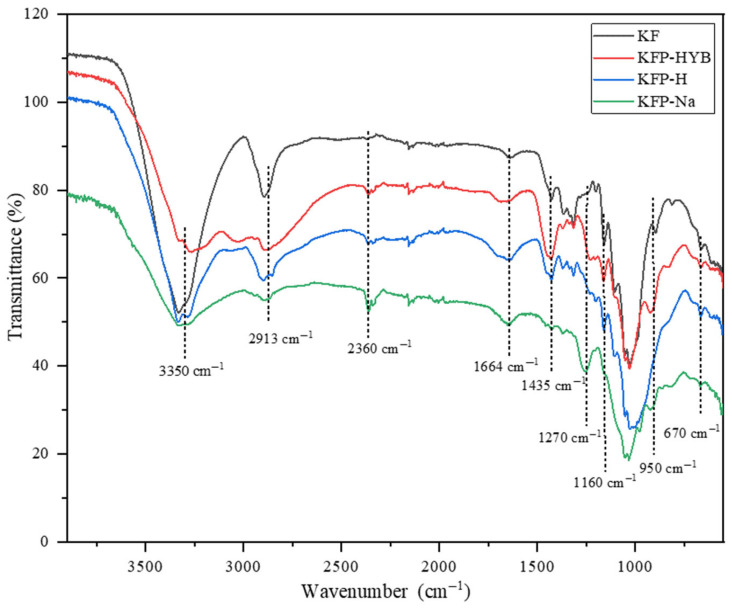
FTIR analysis of KF, KFP-HYB, KFP-H, and KFP-Na.

**Figure 8 polymers-17-02022-f008:**
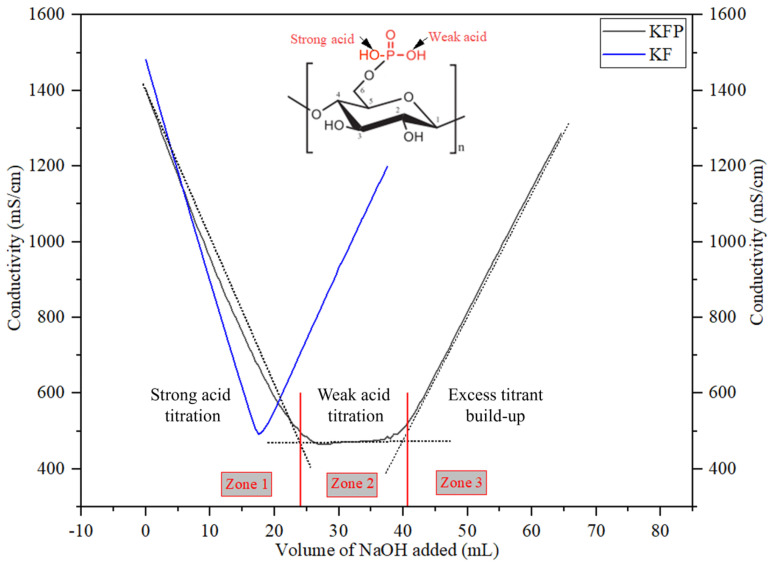
Conductimetric titration of pristine and phosphorylated kraft fibers.

**Figure 9 polymers-17-02022-f009:**
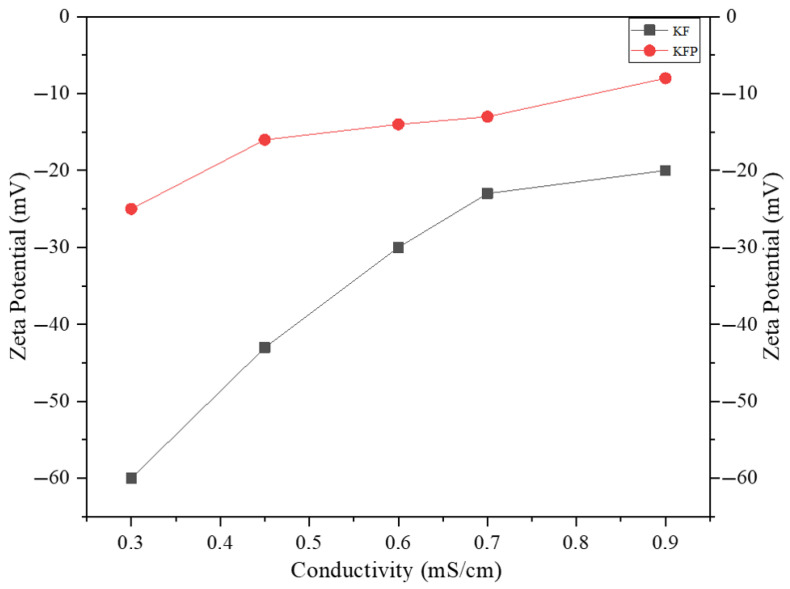
Zeta potential as a function of conductivity for KF and KFP.

**Figure 10 polymers-17-02022-f010:**
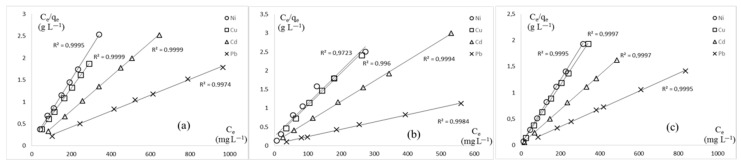
Langmuir linear regressions of adsorption at 20 °C for KFP-HYB (**a**), KFP-H (**b**), and KFP-Na (**c**).

**Figure 11 polymers-17-02022-f011:**
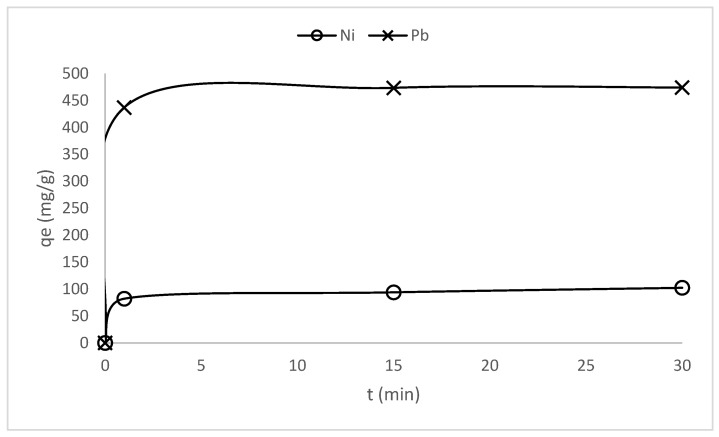
Evolution of the adsorption capacity for nickel and lead onto KFP-HYB fibers.

**Figure 12 polymers-17-02022-f012:**
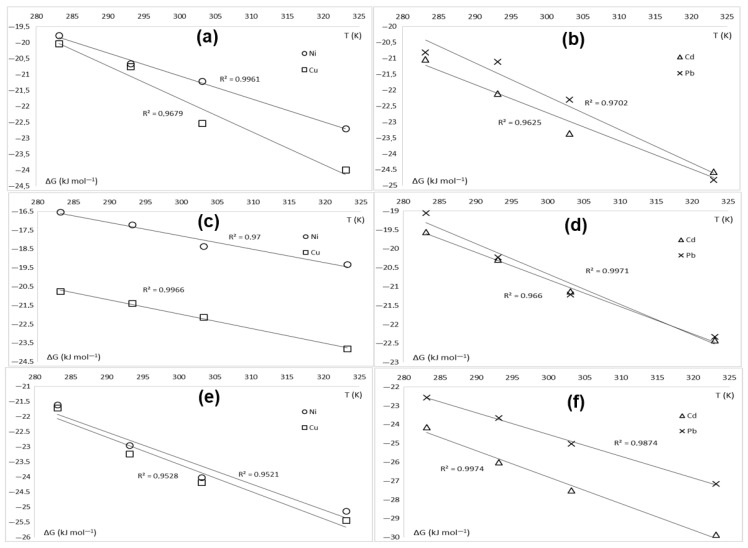
Thermodynamic linear regressions of adsorption for KFP-HYB (**a**,**b**), KFP-H (**c**,**d**), and KFP-Na (**e**,**f**).

**Table 1 polymers-17-02022-t001:** Dimensional and electrostatic properties of KF and PKF.

Fiber Dimensions				Electrostatic Properties in Water			
	Length (mm)	Width (µm)	Fines (%)	Total Charge (mmol/kg)	Surface Charge (µeq/g)	Zeta Potential (mV)	Humidity Content (%)
KF	2.14 ± 0.1	26.6 ± 0.2	0.7 ± 0.1	80 ± 20	30 ± 5	−60 to −20	4 ± 1
KFP	1.95 ± 0.1	32.1 ± 0.2	0.2 ± 0.1	4850 ± 200	530 ± 20	−25 to −8	8 ± 2

**Table 2 polymers-17-02022-t002:** Coefficient of determination for the Langmuir model of adsorption at 20 °C for KFP.

	KFP-HYB	KFP-H	KFP-Na
	R^2^	R^2^	R^2^
Ni	0.9995	0.9723	0.9995
Cu	0.9999	0.9850	0.9997
Cd	0.9999	0.9977	0.9997
Pb	0.9974	0.9984	0.9995

**Table 3 polymers-17-02022-t003:** Comparison of ion exchange capacity at 20 °C for KFP fibers and commercial resins.

	KFP-HYB	KFP-H	KFP-Na	Dowex 50WX2-400	Dowex Marathon C
	*q_max_* (mmol/g)	*q_max_* (mmol/g)	*q_max_* (mmol/g)	* *q_e_* (mmol/g)	* *q_e_ * (mmol/g)
Ni	2.4	1.9	2.8	2.4	2.5
Cu	2.5	2.0	2.8	2.4	2.5
Cd	2.3	1.7	2.7	2.5	2.5
Pb	2.7	2.5	2.9	2.5	2.6

* the values for commercial resins are given at maximum initial concentration for comparison.

**Table 4 polymers-17-02022-t004:** Adsorption capacity of different cellulose-based materials.

Raw Materials	Heavy Metals	pH	Maximum Adsorption Capacity (mmol/g)	References
Aerogel based on amide-functionalized cellulose	Cu(II)	7	0.81	[34]
Aerogel based on nanocellulose and polyethyleneimine	Cu(II) and Pb(II)	2–5	2.74 and 1.73	[35]
Cellulose hydrogels based on acrylamide/acrylic acid	Cu(II), Pb(II), and Cd(II)	1–6	2.48, 1.9, and 2.58	[36]
Carboxymethylated cellulose fibers	Cu(II) and Ni(II)	2–7	0.26 and 0.19	[37]
Cellulose/dibenzo-18-crown 6 composites	Cu(II), Pb(II), Cd, and Ni(II)	6	3.02, 0.97, 1.75, and 3.17	[38]
Magnetic chitosan/cellulose nanofiber-Fe(III)	Pb(II)	1–8	0.48	[39]
KFP-HYB	Cu (II), Pb(II), Cd(II), and Ni(II)	3–10	2.5, 2.7, 2.3, and 2.4	This work
KFP-H			2.0, 2.5, 1.7, and 1.9	This work
KFP-Na			2.8, 2.9, 2.7, and 2.8	This work

**Table 5 polymers-17-02022-t005:** System changes in enthalpy and entropy of adsorption for KFP.

	KFP-HYB		KFP-H		KFP-Na	
	*ΔH* (kJ/mol)	*ΔS* (kJ/mol·K)	*ΔH* (kJ/mol)	*ΔS* (kJ/mol·K)	*ΔH* (kJ/mol)	*ΔS* (kJ/mol·K)
Ni	0.45	0.07	3.50	0.07	2.37	0.09
Cu	9.09	0.10	1.09	0.08	3.38	0.09
Cd	3.17	0.09	0.70	0.07	15.47	0.14
Pb	9.22	0.10	3.41	0.08	10.12	0.12

## Data Availability

The original contributions presented in this study are included in the article. Further inquiries can be directed to the corresponding author(s).

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
