# Peer review of "Thermodynamic Aspects of Ion Exchange Properties of Bio-Resins from Phosphorylated Cellulose Fibers"

_polymers, 2025, doi:10.3390/polym17152022_

Round 1

Reviewer 1 Report

Comments and Suggestions for Authors

Dear Editor,

I have carefully reviewed the manuscript, including its tables and figures, and would like to offer the following recommendation:

The article presents an interesting and timely concept, focusing on the use of waste-derived materials for water purification. The authors describe the modification of cellulose to produce a resin that could potentially replace commercial resins for ion exchange and heavy metal removal. While the topic is relevant and promising, the manuscript requires major revisions before it can be considered for acceptance.

  1. Please find my detailed comments and requests below.
  2. The abstract is too long and should be shortened for clarity and conciseness.
  3. The authors should include zeta potential curves to support their analysis.
  4. The manuscript must present plots showing metal removal as a function of contact time.
  5. The rationale for applying the Langmuir model should be clarified. Did the authors consider or test alternative models, such as the Freundlich isotherm?

Author Response

Question 1: The abstract is too long and should be shortened for clarity and conciseness.

Response 1: The abstract was modified to be clearer and more concise. Please check the revised version of our manuscript from line 10 to line 28.

Question 2: The authors should include zeta potential curves to support their analysis.

Response 2: The zeta potential curves have been added to the manuscript. Please refer to the revised version at lines 360-361.

Question 3: The manuscript must present plots showing metal removal as a function of contact time.

Response 3: The graph showing metal removal as a function of contact time has been added to the revised manuscript, along with a paragraph explaining the results presented in the graph. Please refer to lines 373–381 and lines 387-389.

Question 4: The rationale for applying the Langmuir model should be clarified. Did the authors consider or test alternative models, such as the Freundlich isotherm?

Response 4: Three adsorption models: Langmuir, Freundlich, and Dubinin–Radushkevich, were applied to identify the most suitable model for our application. The determination coefficient (R²) of each model was determined, and the Langmuir model consistently showed the highest fit, with R² values of 0.99 for all tested metals and fiber types. Therefore, the Langmuir model was selected for use in the present study. We have added a paragraph providing more details on this; please refer to lines 361 to 364.

Reviewer 2 Report

Comments and Suggestions for Authors

This article is a very interesting piece of work. The content is detailed and the research methods are reasonable. I think the paper can be published after the minor revisions. Here are my specific suggestions:
1. The length of the abstract is too long. It is recommended to refine the language. For example, remove excessive background information, etc.
2. If there are any reference standards in the Fiber Characterization section, it is also recommended to provide additional explanations.
3. Is the EDX test only conducted in a certain area of the sample? Does the structure of the test exhibit randomness? This is crucial for determining whether the ammonium is replaced by sodium.
4.  Please provide the corresponding references of Langmuir model of adsorption。
5. The font size in the picture is too small. Please enhance the font size and clarity of the picture.
6. It is suggested that key data be included in the conclusion section. For instance, data related to performance improvement. The author seems to have reversed the writing format of the conclusion and the summary. In my opinion, the content of the conclusion should be more comprehensive, while the content of the summary should be more concise.

Author Response

Question 1: The length of the abstract is too long. It is recommended to refine the language. For example, remove excessive background information, etc.

Response 1: The abstract was modified to be clearer and more concise. Please check the revised version of our manuscript from line 10 to line 28.

Question 2: If there are any reference standards in the Fiber Characterization section, it is also recommended to provide additional explanations.

Response 2: Unfortunately, there are no reference standards that could be cited in the Fiber Characterization section. Many of these analyses are developed specifically for the pulp and paper industry. The aim of these empirical analyses is to provide additional information to the industry, to better discriminate between different types of lignocellulosic fibers used in papermaking process.

Question 3:  Is the EDX test only conducted in a certain area of the sample? Does the structure of the test exhibit randomness? This is crucial for determining whether the ammonium is replaced by sodium.

Response 3: EDX analyses were performed in different areas of the samples to obtain a representative overview of the elements present on the fiber surface. It is important to note that EDX only provides information about surface composition. Therefore, the detected percentages of phosphate, nitrogen, and sodium reflect surface concentrations and not bulk values. Typically, UV-Visible titration is used to determine the total content of phosphate or nitrogen within the fibers, as reported in previous studies. However, the focus of this study is primarily on adsorption behavior and thermodynamic analysis. Most fiber characterization data are already available in the external literature and in our studies, some being cited in the References section. In this study, EDX analysis revealed that treating the fibers with NaOH to exchange NH₄⁺ for Na⁺ resulted in a change from a nitrogen content of 5.5% to a sodium content of 6.4%, confirming the complete substitution of NH₄⁺ by Na⁺.

Question 4: Please provide the corresponding references of Langmuir model of adsorption.

Response 4: The reference for the Langmuir adsorption model has been added in the revised manuscript; please refer to line 224.

Question 5:  The font size in the picture is too small. Please enhance the font size and clarity of the picture.

Response 5: All figures have been adjusted to improve clarity and readability for the readers. Please refer to the revised version of the manuscript.

Question 6: It is suggested that key data be included in the conclusion section. For instance, data related to performance improvement. The author seems to have reversed the writing format of the conclusion and the summary. In my opinion, the content of the conclusion should be more comprehensive, while the content of the summary should be more concise.

Response 6: The conclusion has been thoroughly revised, and additional information have been included in this section to highlight the key findings of the study. Please refer to the revised manuscript, lines 464 to 479.

Round 2

Reviewer 1 Report

Comments and Suggestions for Authors

The manuscript can be accepted in the present form.